# Improved Oxygen Uptake Efficiency Parameters Are Not Correlated with VO_2peak_ or Running Economy and Are Not Affected by Omega-3 Fatty Acid Supplementation in Endurance Runners

**DOI:** 10.3390/ijerph192114043

**Published:** 2022-10-28

**Authors:** Zbigniew Jost, Maja Tomczyk, Maciej Chroboczek, Philip C. Calder, Radosław Laskowski

**Affiliations:** 1Department of Biochemistry, Gdansk University of Physical Education and Sport, 80-336 Gdansk, Poland; 2Department of Physiology, Gdansk University of Physical Education and Sport, 80-336 Gdansk, Poland; 3Faculty of Medicine, School of Human Development and Health, University of Southampton, Southampton SO16 6YD, UK; 4NIHR Southampton Biomedical Research Centre, University Hospital Southampton NHS Foundation Trust and University of Southampton, Southampton SO16 6YD, UK

**Keywords:** peak oxygen uptake, oxygen uptake efficiency plateau, running economy, omega-3 fatty acids, endurance runners

## Abstract

Peak oxygen uptake (VO_2peak_) is one of the most reliable parameters of exercise capacity; however, maximum effort is required to achieve this. Therefore, alternative, and repeatable submaximal parameters, such as running economy (RE), are needed. Thus, we evaluated the suitability of oxygen uptake efficiency (OUE), oxygen uptake efficiency plateau (OUEP) and oxygen uptake efficiency at the ventilatory anaerobic threshold (OUE@VAT) as alternatives for VO_2peak_ and RE. Moreover, we evaluated how these parameters are affected by endurance training and supplementation with omega-3 fatty acids. A total of 26 amateur male runners completed a 12-week endurance program combined with omega-3 fatty acid supplementation or medium-chain triglycerides as a placebo. Before and after the intervention, the participants were subjected to a treadmill test to determine VO_2peak_, RE, OUE, OUEP and OUE@VAT. Blood was collected at the same timepoints to determine eicosapentaenoic acid (EPA) and docosahexaenoic acid (DHA) in erythrocytes. OUE correlated moderately or weakly with VO_2peak_ (R^2^ = 0.338, *p* = 0.002) and (R^2^ = 0.226, *p* = 0.014) before and after the intervention, respectively. There was a weak or no correlation between OUEP, OUE@VAT, VO_2peak_ and RE despite steeper OUE, increased OUEP and OUE@VAT values in all participants. OUE parameters cannot be treated as alternative parameters for VO_2peak_ or RE and did not show changes following supplementation with omega-3 fatty acids in male amateur endurance runners.

## 1. Introduction

There are many cardiopulmonary exercise tests (CPETs) that aim to assess parameters related to human performance, such as peak oxygen uptake (VO_2peak_) or maximal oxygen uptake (VO_2max_). VO_2max_ is considered the best indicator of potential in endurance events, being a ‘gold standard’ measurement of integrated cardiopulmonary-muscle oxidative function [1,2,3]. Although heart rate (HR), respiratory exchange ratio (RER), and minute ventilation (Ve) are considered cardiovascular, respiratory, and pulmonary parameters, respectively, their comprehensive function is often difficult to evaluate. Therefore, there is a need to identify alternative validated and reliable parameters for assessing cardiorespiratory fitness.

Sun and co-authors [4] determined the relationship between oxygen uptake (VO_2_) and Ve, called oxygen uptake efficiency (OUE). They noted that OUE increases linearly with time during early exercise, but becomes non-linear as Ve increases faster than VO_2_. This curvilinear relationship during an exercise test is not as appropriate for assessing aerobic capacity as VO_2peak_. Thus, the authors described other physiological parameters that can be determined from respiratory gases during CPET, i.e., oxygen uptake efficiency at the ventilatory anaerobic threshold (OUE@VAT) and oxygen uptake efficiency plateau (OUEP) in healthy subjects. It was observed that both OUE@VAT and OUEP are simple measurements that do not require maximum effort. Moreover, they are also easy to visualise, recognise and calculate [4], making them potentially robust parameters for assessing physical fitness. It is worth noting that there is still scarce evidence of improvements in OUE parameters after physical training and no evidence of improvements after supplementation with bioactive compounds such as the omega-3 fatty acids (eicosapentaenoic acid (EPA) and docosahexaenoic acid (DHA)).

Although some studies show no improvements in cardiopulmonary-muscle oxidative function following supplementation with fish oil containing omega-3 fatty acids [5,6], several studies do indicate a positive effect. For example, long-term EPA and DHA supplementation may contribute to the improvement in VO_2max_ [7] or to the reduction in the cost of aerobic exercise in trained cyclists [8,9]. Moreover, our recent study showed that 12-week supplementation with omega-3 fatty acids improved running economy (RE) in amateur runners [10]. These studies focus mainly on VO_2max_ and RE, and other submaximal oxygen kinetics parameters need to be further explored.

The aim of our study was to determine whether OUE, OUEP and OUE@VAT can be considered as a robust measurements of endurance capacity. Moreover, we verify if those parameters are sensitive to changes after omega-3 fatty acid supplementation. The main hypothesis of this research was that OUE will be sensitive to changes in VO_2peak_. We also hypothesised that OUE@VAT and OUEP can be used as non-invasive, submaximal parameters of oxygen kinetics replacing VO_2peak_ and RE. We also evaluated whether twelve-week of endurance training combined with omega-3 fatty acid supplementation can alter these parameters in male amateur endurance runners.

## 2. Materials and Methods

### 2.1. Participants

A total of 26 male amateur runners (37 ± 3 years old; 77 ± 9 kg body weight; VO_2peak_: 54.2 ± 6 mL*kg^−1^*min^−1)^ completed the 12-week experimental study as previously described [10], which tested the effect of supplementation with omega-3 fatty acids on exercise capacity in male amateur endurance runners. Participants were not taking medication and all were in good health, as confirmed by a medical check. The study was approved by the Bioethical Committee of Regional Medical Society in Gdańsk (NKBBN/628/2019) and conducted according to the Declaration of Helsinki (2013). All participants provided their written informed consent prior participating in the study. Detailed participant characteristics and project design are shown in Table 1 and Figure 1, respectively.

### 2.2. Supplementation

Participants were randomly assigned to one of two groups with the final characteristics as follows: OMEGA (37 ± 3 years; 76.3 ± 11 kg body weight; VO_2peak_: 53.6 ± 4 mL*kg^−1^*min^−1^) or medium-chain triglycerides (MCT) as placebo (37 ± 4 years; 78 ± 8 kg body weight; VO_2peak_: 54.7 ± 7 mL*kg^−1^*min^−1^). The division of participants into two groups was performed to check the difference in the response of OUE parameters to supplementation. Hence, participants supplemented four capsules per day, providing a total of 2234 mg of EPA + 916 mg of DHA (OMEGA group) or 4000 mg of MCT (MCT group). The capsules were provided in coded, identical-looking packages to avoid a potential recognition. To maintain the quality of supplements consisting of omega-3 fatty acids and their respective dosages, materials adhering to the International Fish Oil Standard (IFOS) were used.

### 2.3. Treadmill Exercise Testing

Exercise tests were conducted under controlled environmental conditions (18–20 °C and humidity 40–45%) and were performed at similar time of day ± 2 h. Before carrying out the exercise tests, the participants performed a familiarization trial. The participants were informed to refrain from strenuous exercise for 24 h and from caffeine or alcohol consumption for 12 h prior to the tests. Before and after twelve weeks of the training program, participants undertook a ramp exercise test to volitional exhaustion on a treadmill (*h*/*p* Cosmos, Saturn, Nussdorf-Traunstein, Germany). First, participants stood on the treadmill for 2 min to make sure the measuring equipment was ready and to measure the resting parameters. Thereafter, runners walked for 5 min at 5 km/h speed and with a 1.5% inclination as a warm-up prior to starting the test. Every next stage lasted 3 min, and the treadmill belt was accelerated starting from 8 km/h by 1 km/h per stage up to 12 km/h. Then, the inclination of the treadmill was increased to 5%, 10% and 15% at 12 km/h speed until volitional exhaustion, despite strong verbal encouragement. During both tests, heart rate (HR) was monitored (Polar RS400, Kempele, Finland). RE was measured as an oxygen cost from last 50 s of each stage to 12 km/h speed and was expressed as mL*kg^−^^1^*min^−^^1^ [11].

### 2.4. Respiratory Gas Measurements

During both laboratory tests, the exhaled air was continuously measured using a breath-by-breath analyser (Oxycon Pro, Jaeger, Hoechberg, Germany). Before the tests, the analyser was calibrated in accordance with the manufacturer’s instructions. All measurements were averaged to 10 s intervals and included: oxygen uptake (VO_2_), carbon dioxide output (VCO_2_), minute ventilation (Ve), end-tidal pressure of oxygen (P_ET_O_2_) and end-tidal pressure of carbon dioxide (P_ET_CO_2_).

### 2.5. Determination of Oxygen Uptake Efficiency and Ventilatory Thresholds

The OUE was individually determined for each participant by calculating the regression slope from the linear relationship of absolute VO_2_ (mL*min^−1^) plotted as a linear function of Ve (L*min^−1^) (VO_2_ = Ve + b), as previously described by Sun et al. [4]. After calculating the OUE individually for each participant from the formula, the OUE was correlated with the true VO_2peak_ and normalized, and the original OUE values (“b”) were compared for the slope of the linear regression of the oxygen uptake efficiency. OUEP was calculated as the 90 s average of the highest consecutive measurements of VO_2_ (mL*min^−1^)/Ve (L*min^−1^) and OUE at the ventilatory anaerobic threshold (OUE@VAT), as the 60 s average of consecutive measurements at and immediately before the VAT accordingly to Sun et al. [4]. First, ventilatory threshold (VT_1_) was determined as increase in both the ventilatory equivalent of oxygen (Ve/VO_2_) and end-tidal pressure of oxygen (P_ET_O_2_) with no concomitant increase in the ventilatory equivalent of carbon dioxide (Ve/VCO_2_) [12]. The ventilatory anaerobic threshold (VAT) was measured by the V-slope method [13]. Peak oxygen uptake (VO_2peak_) was obtained as the last 30 s oxygen uptake mean value recorded during the test [14].

### 2.6. Training Program

All participants underwent 12 weeks of an endurance training program. The participants performed endurance training of varying intensity three times a week according to Costa et al. [15] with slight modifications. Additionally, participants performed training once a week, which aimed to strengthen the central stabilization muscles and to reduce the risk of injury [16]. The training intensity was distributed among 3 heart-rate zones (Z1-Z2-Z3). They were determined according to the first ventilatory threshold (VT_1_), ventilatory anaerobic threshold (VAT) and the corresponding values of the heart rate [Z1: ≤HR@VT_1_ + 5 bpm; Z2: (>HR@VT_1_ + 5 bpm) to (≤HR@VAT-5 bpm); Z3: >HR@VAT-5 bpm]. Average training times spent in every mesocycle were (~80%-15%-5%) in zones (Z1-Z2-Z3), respectively. In the last week, the training volume was reduced to reduce the accumulated fatigue. All trainings were monitored by Polar M430 (Kempele, Finland) wrist watches and H9 heart-rate chest sensor and the supervision over the participants was carried out by a certified track and field coach.

### 2.7. Erythrocyte Fatty Acid Analysis

Sample collection and fatty acid determination were outlined elsewhere [10]. In brief, blood samples were collected into 4 mL sodium citrate vacutainer tubes and centrifuged at 4 °C (4000× *g* for 10 min). After centrifugation, plasma was collected with a disposable Pasteur pipette, transferred into separate Eppendorf probes and stored in a −80 °C freezer until further analysis. Erythrocyte lipids were extracted into chloroform:methanol and fatty acid methyl esters (representing the erythrocyte fatty acids) were formed by heating the lipid extract with methanolic sulphuric acid. The fatty acid methyl esters were separated by gas chromatography on a Hewlett Packard 6890 gas chromatograph fitted with a BPX-70 column using the settings and run conditions described by Fisk et al. [17]. Fatty acid methyl esters were identified by comparison with runtimes of authentic standards and data were expressed as weight % of total fatty acids.

### 2.8. Statistical Analysis

Statistical analysis was performed using GraphPad Prism 7 (San Diego, CA, USA). Arithmetic means, standard deviation (SD), and significance levels of differences between means were calculated. Two-way analysis of variance (ANOVA), with repeated measures, was used to investigate the significance of differences between groups and time. Significant main effects were further analyzed using the Bonferroni corrected post hoc test. Correlations between variables were evaluated using the Pearson and Spearman correlations coefficients. All analyses used a significance level of *p* < 0.05.

## 3. Results

### 3.1. Predicted VO_2peak_ from OUE Equation

Predicted VO_2peak_ calculated from the OUE formula both before and after the supplementation intervention was moderately correlated with peak oxygen uptake (R^2^ = 0.338, *p* = 0.002; Figure 2A) for all participants before the study. Moreover, the results without grouping also showed a correlation after 12 weeks of intervention (R^2^ = 0.226, *p* = 0.014; Figure 2B), but the correlation was weak.

### 3.2. Oxygen Uptake Efficiency Plateau

Pre-intervention OUEP values weakly correlated with VO_2peak_ (R^2^ = 0.247, *p* = 0.01; Figure 3A). After twelve weeks of intervention, no correlation was found between these two indicators (R^2^ = 0.077, *p* = 0.17, Figure 3B).

### 3.3. Oxygen Uptake Efficiency at the Ventilatory Anaerobic Threshold

OUE@VAT poorly correlated with the peak oxygen uptake (VO_2peak_) before the study (R^2^ = 0.179, *p* = 0.031, Figure 4A) and there was no correlation after the 12-week intervention (R^2^ = 0.082, *p* = 0.154, Figure 4B) in all participants.

### 3.4. Correlation between OUEP, OUE@VAT and RE

The changes observed in RE (presented as VO_2_ delta [%] at 12 km/h) were not correlated with the change in OUEP (R^2^ = 0.018, *p* = 0.511; Figure 5A). Similar results were observed in the correlation between RE and OUE@VAT (r = 0.079, *p* = 0.699; Figure 5B) in all participants.

### 3.5. Omega-3 Fatty Acids Supplementation

Baseline levels of EPA and DHA did not differ between the groups (OMEGA group: 1.1% EPA, 4.7% DHA; MCT group: 1.2% EPA, 4.4% DHA, both *p* > 0.999). Post-intervention values of EPA and DHA increased in OMEGA group (4.9% EPA, 6.7% DHA, both *p* < 0.001). Changes were not observed in MCT group (1.2% EPA, *p* > 0.999; 4.7% DHA, *p* = 0.551). All results are provided in Table 1.

#### 3.5.1. Oxygen Uptake Efficiency

At the end of the 12-week supplementation period, there was an increase in the slope of oxygen uptake efficiency in the OMEGA group from 35.4 ± 3.3 to 37.6 ± 3.0 and in the MCT group from 35.5 ± 3.7 to 37.2 ± 3.1; (both *p* < 0.001). OUE increased when groups were combined from 35.5 ± 3.4 to 37.4 ± 3.0; (*p* < 0.001, Table 2).

#### 3.5.2. Oxygen Uptake Efficiency Plateau

Oxygen uptake efficiency plateau values increased in the OMEGA group from 41.3 ± 4.6 to 43.6 ± 4.0; (*p* = 0.017). There were no changes in the MCT group (*p* = 0.2). Moreover, the analysis of the two groups together (regardless of the supplementation that was undertaken) showed that OUEP increased from 41.6 ± 4.8 to 43.2 ± 3.9; (*p* = 0.007, Table 2).

#### 3.5.3. Oxygen Uptake at Ventilatory Anaerobic Threshold

There was an increase in OUE@VAT in the OMEGA group from 32.7 ± 3.6 to 35.9 ± 4.7; (*p* = 0.012) and in the MCT group from 33.2 ± 3.8 to 35.4 ± 3.5; (*p* = 0.003). The results, regardless of the supplementation undertaken, showed that OUE@VAT increased from 32.9 ± 3.7 to 35.7 ± 4.1; (*p* < 0.001, Table 2).

## 4. Discussion

This is the first study to report the correlations between OUE, OUEP, OUE@VAT and VO_2peak_ as well as OUEP and OUE@VAT and RE. They were analyzed in terms of reliability and repeatability, and whether they could be non-invasive substitute measurements for VO_2peak_ and RE. Additionally, we investigated whether these parameters were altered following supplementation with omega-3 fatty acids.

The true VO_2max_ value is mainly achievable during a laboratory progressive exercise test to exhaustion where large muscle groups are involved. Simultaneously, the observed kinetics of oxygen supply/utilization in the muscles must be without significant changes: the so-called plateau [18]. It is known that this phenomenon occurs when a high intensity is met, and the primary criteria for achieving this parameter (VO_2max_) during CPET are: (1) reaching a VO_2_ plateau or (2) levelling-off the oxygen uptake (VO_2_) [19,20,21]. Thus, in Sun and co-authors’ study, OUE, OUEP and OUE@VAT comprehensively reflected cardiovascular functions as an alternative for parameters assessing CRF without the need for maximum effort [4]. A steeper OUE (VO_2_/Ve) and higher values of OUEP and OUE@VAT show more efficient oxygen uptake and utilization in the working skeletal muscles. OUE showed an improvement, but, for both groups, this occurred after 12 weeks of intervention. Hence, it is believed that the increase in slope/higher OUE values was the result of endurance training. Moreover, in our study, weak or no correlation was observed between OUEP and peak oxygen uptake. In a study by Bongers et al. [22], in which 214 children participated, OUEP was weak-to-moderately correlated with VO_2peak_ (r = 0.646), which is inconsistent with our results. However, children and adults respond differently to exercise, which might explain this difference. Another study also confirms that OUEP does not accurately predict VO_2max_ in male adolescents and should not replace VO_2max_, when assessing CRF [19]. In our study, OUE@VAT also demonstrated no correlation with VO_2peak_ before and after 12 weeks of intervention. In contrast to our results, one study revealed that ventilatory anaerobic threshold (VAT) strongly correlated with VO_2peak_ (r = 0.831) [23]. However, there is a difference between the compared parameters, because OUE@VAT is the 60-s average of consecutive measurements at and immediately before the VAT. On the other hand, VAT is a single measurement and is not free from intra-observer and inter-observer variability [24]. Hence, both OUEP and OUE@VAT may be more stable measurements than VAT; however, the results of our study did not confirm this.

Endurance capacity also has a stable predictor in the form of RE [25]. However, as earlier authors suggest, an accurate measurement of RE can be carried out with the use of invasive lactate measurement, which is one of the disturbances in VO_2_ steady-state indicators [26,27]. Therefore, in this study, an attempt was made to replace RE with OUEP and OUE@VAT and to check whether they can be a solid, non-invasive predictor of RE in recreational runners. Despite the increase in the efficiency of oxygen uptake in all participants, the linear regression did not show any correlation between OUEP, OUE@VAT and RE. Hence, the RE measurement should not be replaced with OUEP and OUE@VAT, as they are not related.

The assessment of adaptive changes following supplementation with omega-3 fatty acids is also not fully known. The health-promoting effects of n-3 PUFA supplementation are well-established [28,29,30]. These effects are related to the incorporation of EPA and DHA into the erythrocyte cell membrane [31], skeletal muscles [32] and heart [33]. Furthermore, the systemic response to supplementation with omega-3 fatty acids as exemplified by maximum oxygen uptake [7], exercise economy [9,10] or anaerobic endurance capacity [34] is well-known. Nevertheless, in our study, for the first time, an attempt was made to link the effect of supplemental EPA + DHA to changes in OUEP and OUE@VAT. However, the OUE parameters increased in both groups. Therefore, changes in OUEP and OUE@VAT following 12 weeks of intervention are dictated by adaptation to endurance training rather than changes caused by EPA and DHA supplementation.

### Limitations and Future Perspectives

Despite some valuable information coming from this study, there are some limitations. First, the small number of participants could distort the estimate of correlations between the variables. Second, this study was conducted in male runners only; therefore, these findings cannot be generalized and extrapolated to females. Future studies should include a larger number of participants and include females.

## 5. Conclusions

In conclusion, the results obtained in this study do not support the use of OUEP and OUE@VAT as an alternative parameter to VO_2peak_ and RE. Additionally, the 12-week supplementation of omega-3 fatty acids at a dose of 2234 mg of EPA and 916 mg of DHA daily did not reveal changes in OUEP and OUE@VAT. Hence, the suitability of using OUEP and OUE@VAT as alternative, non-invasive CRF parameters for VO_2peak_ and RE can be questioned.

## Figures and Tables

**Figure 1 ijerph-19-14043-f001:**
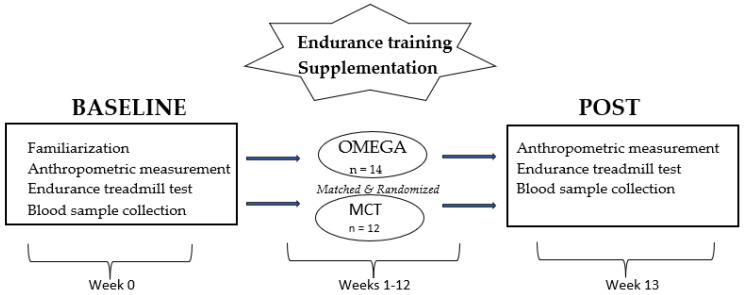
Procedure design.

**Figure 2 ijerph-19-14043-f002:**
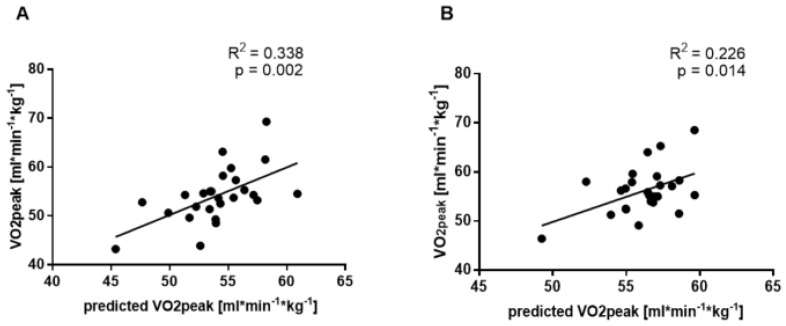
The linear relationship between VO_2peak_ and predicted VO_2peak_ before (**A**) and after (**B**) twelve weeks of combined endurance training and supplementation (OMEGA and MCT groups; n = 26).

**Figure 3 ijerph-19-14043-f003:**
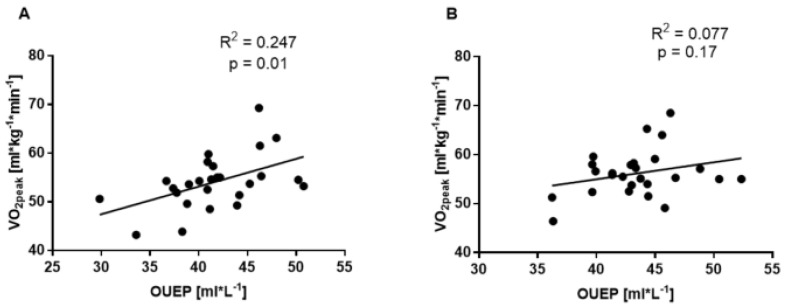
The linear relationship between VO_2peak_ and OUEP before (**A**) and after (**B**) twelve weeks of combined endurance training and supplementation (OMEGA and MCT groups; n = 26).

**Figure 4 ijerph-19-14043-f004:**
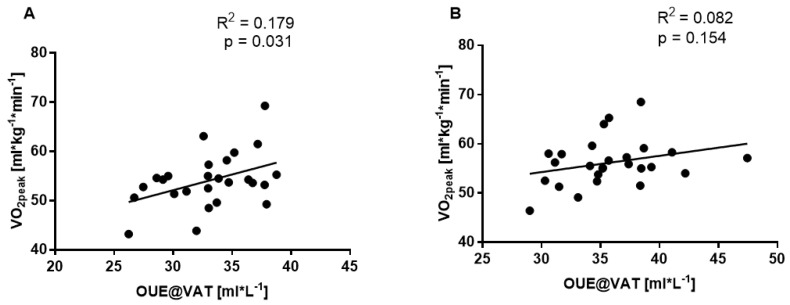
The linear relationship between VO_2peak_ and OUE@VAT before (**A**) and after (**B**) twelve weeks of combined endurance training and supplementation (OMEGA and MCT groups; n = 26).

**Figure 5 ijerph-19-14043-f005:**
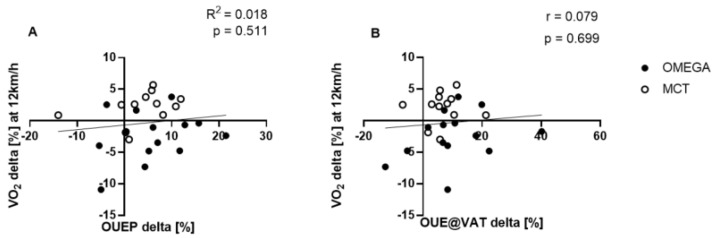
Correlation between changes in RE and OUEP (**A**) and OUE@VAT (**B**) after twelve weeks of combined endurance training and supplementation (OMEGA and MCT groups; n = 26).

**Table 1 ijerph-19-14043-t001:** Characteristics of participants.

Variable		MCT (n = 12) Mean ± SD	OMEGA (n = 14) Mean ± SD
Age [y]		37 ± 4	37 ± 3
Body mass [kg]		78.0 ± 8	76.3 ± 11
Height [cm]		180 ± 4	181 ± 7
EPA [% of total erythrocyte fatty acids]	Pre	1.2 ± 0.3	1.1 ± 0.4
	Post	1.2 ± 0.3	4.9 ± 1.1 *^,^^
DHA [% of total erythrocyte fatty acids]	Pre	4.4 ± 1.1	4.7 ± 1.0
	Post	4.5 ± 0.8	6.7 ± 0.8 *^,^^
HRmax [beats*min^−1^]	Pre	186 ± 9	190 ± 9
	Post	184 ± 7	189 ± 9
VO_2peak_ [mL*kg^−1^*min^−1^]	Pre	54.7 ± 6.8	53.6 ± 4.4
	Post	56.4 ± 5.9	56.0 ± 3.7 *
RE [mL*kg^−1^*min^−1^]	Pre	47.7 ± 3.3	47.6 ± 1.8
	Post	48.7 ± 2.9	46.5 ± 2.4 ^^^

EPA—eicosapentaenoic acid; DHA—docosahexaenoic acid; HRmax—maximal heart rate; RE—running economy; data are presented as mean ± SD; * *p* < 0.05 for post vs. pre value ^^^
*p* < 0.05 for MCT vs. OMEGA.

**Table 2 ijerph-19-14043-t002:** Comparison of effects omega-3 fatty acid supplementation with placebo controlled on cardiorespiratory fitness (CRF) parameters.

Variable	MCT (n = 12) Mean ± SD		OMEGA (n = 14) Mean ± SD		ALL (n = 26) Mean ± SD	
	Pre	Post	Pre	Post	Pre	Post
OUE [mL*L^−1^]	35.5 ± 3.7	37.2 ± 3.1 ***	35.4 ± 3.3	37.6 ± 3.1 ***	35.5 ± 3.4	37.4 ± 3.0 ***
OUEP [mL*L^−1^]	41.8 ± 5.2	42.9 ± 3.8	41.3 ± 4.6	43.6 ± 4.0 *	41.6 ± 4.8	43.2 ± 3.9 **
OUE@VAT [mL*L^−1^]	33.2 ± 3.8	35.4 ± 3.5 **	32.7 ± 3.6	35.9 ± 4.7 *	32.9 ± 3.7	35.7 ± 4.1 ***
Ve [L*min^−1^]	93.8 ± 11.6	90.7 ± 9.3 **	92.9 ± 20.4	87.4 ± 20.2 *	93.3 ± 16.4	88.9 ± 15.7 *

OUE—oxygen uptake efficiency; OUEP—oxygen uptake efficiency plateau; OUE@VAT—oxygen uptake efficiency at the ventilatory anaerobic threshold; Ve—minute ventilation; * *p* < 0.05 for post to pre value; ** *p* < 0.01 for post to pre value; *** *p* < 0.001 for post to pre value; data are presented as mean ± standard deviation (SD).

## Data Availability

The data presented in this study are available on request from the corresponding author.

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
