# Peer review of "Improved Oxygen Uptake Efficiency Parameters Are Not Correlated with VO2peak or Running Economy and Are Not Affected by Omega-3 Fatty Acid Supplementation in Endurance Runners"

_ijerph, 2022, doi:10.3390/ijerph192114043_

Round 1

Reviewer 1 Report

First of all, I would like to thank you for the opportunity given to review the following research. It is undoubtedly a research that shows a very good methodological consistency, however, I believe that it should be improved in the aspects indicated below: 

In the final section of the introduction add research objectives, as they will complement the research hypotheses. 

Regarding the material and method, the following sections should be improved: 

Indicate the type of research that has been carried out: for example, indicate whether the study is experimental or non-experimental, cross-sectional or longitudinal, etc. 

Also, create a new section entitled Instruments and Variables, indicating the variable under study and the instrument used to collect that variable. I think this will make everything more orderly. 

Add a section entitled "Procedure" where the steps taken from the beginning of the research until the end of the research are mentioned. 

Add a new section entitled "Limitation and Future Perspectives" where the limitations and future lines of study arising from this research are discussed. 

Finally, from line 306 to 313 complete the Author Contributions. 

Author Response

Thank you for your suggestions and comments. Our responses are available in the attachment.

Reviewer 2 Report

The main aim of the study „Improved oxygen uptake efficiency parameters are not correlated with VO2peak or running economy and are not affected by omega-3 fatty acid supplementation in endurance runners“ is to examine if OUE will be sensitive to changes in VO2peak. The authors also hypothesised that OUE@VAT and OUEP can be used as non-invasive, submaximal parameters of oxygen kinetics replacing VO2peak and RE. And they also evaluated whether twelve-week of endurance training combined with omega-3 fatty acid supplementation can alter these parameters in male amateur endurance runners.

The study is original and interesting. But it is complicated and at first glance not quite clearly described. I would like to appretiate the efforts of the authors. However, some information is missing in the article, some facts need to be explained and add:

In the introduction, I miss more substantiated information about the influence of the selected substances on the selected parameters and better presentation of the issue addressed in the study.

The design of the study is described vaguely to the unbiased reader, some information is not obvious at first glance. I miss in the methodological part an explanation of why these two groups (OMEGA and MCT) were chosen, what the meaning of these groups and the substances used (placebo?). The concept of placebo is not mentioned in the methodological part. There is only a hint of an explanation in the “supplementation” part.

Does it say anywhere whether the substance chosen for supplementation has an effect on VO2peak and RE? In the previous study, this information is available, but I did not find it in this study (if I missed it, it should be stressed). If an alternative evaluation to VO2peak and RE is identified, information on the effect of the substance on VO2peak and RE should be provided in the methodology part.

Line: 72: Study is previously described (10) – i recommend very briefly writing what the study was about.

I recommend better formulating the conclusion.

Author Response

(The authors gave the same response as above.)

Round 2

Reviewer 2 Report

My comments have been incorporated, questions explained, I have no further comments. Thank you.